# VisualWebBench: How Far Have Multimodal LLMs Evolved in Web Page Understanding and Grounding?

**Junpeng Liu**[♣,*] **Yifan Song**[○,*] **Bill Yuchen Lin**[§] **Wai Lam**[♣] **Graham Neubig**[♠]
**Yuanzhi Li**[◇] **Xiang Yue**[♠]

♠ Carnegie Mellon University
♣ The Chinese University of Hong Kong
○ School of Computer Science, Peking University
◇ MBZUAI   § Allen Institute for AI

https://visualwebbench.github.io/

## Abstract

Multimodal Large Language models (MLLMs) have shown promise in web-related tasks, but evaluating their performance in the web domain remains a challenge due to the lack of comprehensive benchmarks. Existing benchmarks are either designed for general multimodal tasks, failing to capture the unique characteristics of web pages, or focus on end-to-end web agent tasks, unable to measure fine-grained abilities such as OCR, understanding, and grounding. In this paper, we introduce VisualWebBench, a multimodal benchmark designed to assess the capabilities of MLLMs across a variety of web tasks. VisualWebBench consists of seven tasks, and comprises 1.5K human-curated instances from 139 real websites, covering 87 sub-domains. We evaluate 16 open-source MLLMs, Gemini Pro, Claude-3 series, and GPT-4V(ision) on VisualWebBench, revealing significant challenges and performance gaps. Further analysis highlights the limitations of current MLLMs, including inadequate grounding in text-rich environments and subpar performance with low-resolution image inputs. We believe VisualWebBench will serve as a valuable resource for the research community and contribute to the creation of more powerful and versatile MLLMs for web-related applications.

## 1 Introduction

The web is an indispensable platform for information exchange and interaction, presenting unique challenges and opportunities for multimodal learning. While web content has been a primary source of training data for multimodal large language models (MLLMs) (OpenAI, 2023; Google et al., 2023; Liu et al., 2023a), a largely overlooked aspect is understanding of websites themselves. Every website is designed to be visually rendered for consumption by human users, with structured layouts, rich textual information, and diverse interactive elements. Enabling MLLMs to accurately comprehend websites would unlock numerous applications in the web domain.

However, evaluating the performance of multimodal models in the web domain is a challenging task. Unlike object- or scene-centric images in most existing benchmarks (Young et al., 2014; Goyal et al., 2017; Lin et al., 2014; Singh et al., 2019; Li et al., 2023b; Liu et al., 2023b; Yu et al., 2023; Yue et al., 2023), web pages present a complex interplay of visual and textual information, along with interactive elements, requiring models to possess rigorous understanding abilities over hierarchical structures and contextual relationships. Moreover, web elements are often small, numerous, and scattered across the page, demanding

---

[*]Equal contribution.
[†]Corresponding to: jpliu@link.cuhk.edu.hk   xyue2@andrew.cmu.edu.

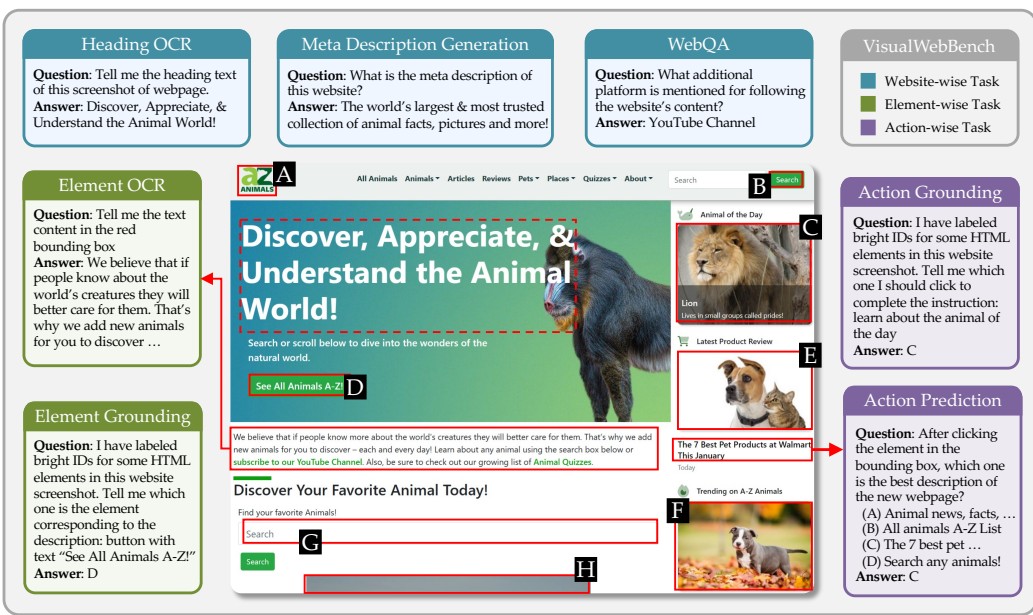

Figure 1: `VisualWebBench` contains seven QA-style tasks, covering website, element, action-level understanding, reasoning, and grounding capabilities.

fine-grained recognition and accurate spatial reasoning and grounding. The vast diversity of website designs, layouts, and content across different domains further complicates the creation of representative and robust evaluation benchmarks, necessitating the inclusion of a wide range of website categories to ensure the generalizability of the evaluated models.

Despite the growing importance of the web domain in multimodal learning, existing benchmarks fall short of comprehensively evaluating the fundamental capabilities of models in this context. General MLLM benchmarks (Young et al., 2014; Liu et al., 2023b; Yue et al., 2023), do not adequately capture the unique characteristics of the web domain. On the other hand, web-agent benchmarks, like WebShop (Yao et al., 2022), Mind2Web (Deng et al., 2024), and (Visual)WebArena (Zhou et al., 2023; Koh et al., 2024), focus on end-to-end abilities without offering a fine-grained assessment of essential skills such as OCR, semantic understanding, and grounding. Measuring these fine-grained abilities is crucial, as they serve as building blocks for complex web-related tasks, enable targeted improvements, and provide a clearer picture of a model's performance. The lack of granularity in existing benchmarks hinders the development of more capable multimodal models for the web domain, emphasizing the need for a comprehensive evaluation benchmark.

To address these limitations, we introduce `VisualWebBench`, a comprehensive multimodal benchmark designed to assess the capabilities of MLLMs in the web domain. Inspired by the human interaction process with web browsers, `VisualWebBench` consists of seven tasks that map to core abilities required for web tasks: meta description generation, webpage QA, heading OCR, element OCR, element grounding, action prediction, and action grounding, as detailed in Figure 1. The benchmark comprises 1.5K instances, all uniformly formulated in the QA style, making it easy to evaluate and compare the performance of different MLLMs.

We evaluate 16 open-source MLLMs, Gemini Pro (Google et al., 2023), Claude Sonnet, Claude Opus (Anthropic, 2024), and GPT-4V(ision) (OpenAI, 2023) on `VisualWebBench`; our key findings are as follows:

- `VisualWebBench` presents significant challenges for current MLLMs, with GPT-4V and Claude Sonnet achieving average scores of 64.6 and 65.8, respectively, indicating substantial room for improvement.

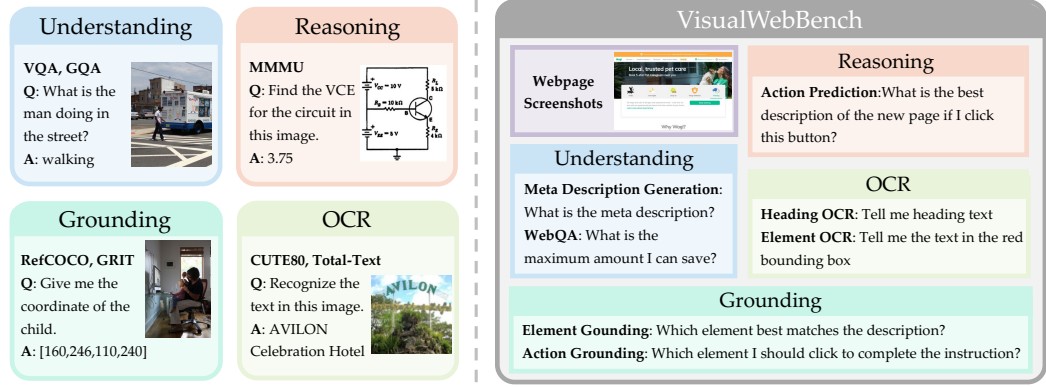

Figure 2: Comparison between `VisualWebBench` (right) and other multimodal benchmarks (left).

- A notable performance gap exists between open-source MLLMs and proprietary counterparts such as GPT-4V and Claude series, with the leading open-source model, LLaVA-1.6-34B, achieving an average score of 50.5.
- MLLMs' abilities in general domains, such as general reasoning on MMMU (Yue et al., 2023), and web agent tasks, such as Mind2Web (Deng et al., 2024), do not correlate much with their performance on `VisualWebBench`, highlighting the importance of web-specific benchmarks like `VisualWebBench`.
- The limited image resolution handling capabilities of most open-source MLLMs restrict their utility in web scenarios, where rich text and elements are prevalent.
- Grounding ability, a crucial skill for developing MLLM-based web applications like autonomous web agents, is a weakness for most MLLMs.

In summary, `VisualWebBench` offers a standardized benchmark for evaluating MLLMs in web understanding, enabling the development of more capable and efficient models, autonomous web agents, and web-related applications.

## 2 Related Work

Before detailing `VisualWebBench`, we briefly outline its differences with existing MLLM benchmarks, also outlined in Figure 2.

### 2.1 MLLM Benchmarks

In concert with improvements in these MLLMs, benchmarks have also evolved. These range from traditional single task benchmarks like VQA (Antol et al., 2015; Goyal et al., 2017), RefCOCO (Mao et al., 2016), and Flickr30K (Young et al., 2014), to more holistic evaluation benchmarks like LAMM (Yin et al., 2024), MMBench (Liu et al., 2023b), and MMMU (Yue et al., 2023), recently. In this work, we focus on images in web-based scenarios characterized by structured layouts, copious textual data, and diverse interactive elements, which pose new challenges for current MLLMs. The most closely related scenario to this work is GUI-based tasks, exemplified by Screen2Words (Wang et al., 2021), Widget Captioning (Li et al., 2020), and WebSRC (Chen et al., 2021) which is a web-based VQA dataset. Different from previous works, `VisualWebBench` offers a comprehensive evaluation for MLLMs, spanning perception, comprehension, grounding, and reasoning capabilities.

### 2.2 Web Agent Benchmarks

As a vital aspect of daily life, methods that perform various tasks in web scenarios have garnered widespread attention from researchers. Earlier efforts introduce simplified sim-

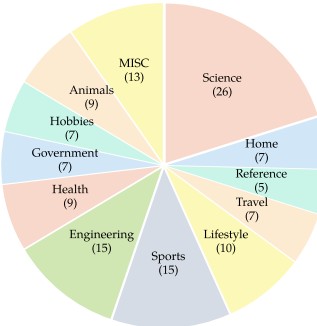

| Task | Level | Capability | Metric | #Num. |
|------|-------|-----------|--------|-------|
| Meta Description Generation | | Understanding | ROUGE-L | 134 |
| WebQA | Webpage | Understanding | F1 | 314 |
| Heading OCR | | OCR | ROUGE-L | 46 |
| Element OCR | Element | OCR | ROUGE-L | 245 |
| Element Grounding | | Grounding | Accuracy | 413 |
| Action Prediction | Action | Reasoning | Accuracy | 281 |
| Action Grounding | | Grounding | Accuracy | 103 |
| Total | - | - | - | 1534 |

Figure 3: Overview of `VisualWebBench`. Left: Domain distribution. The numbers represent the count of sub-domains within each domain. Right: Tasks in `VisualWebBench`.

ulated environments for web navigation tasks, such as MiniWob++ (Liu et al., 2018) and WebShop (Yao et al., 2022). Recently, Mind2Web (Deng et al., 2024), WebArena (Zhou et al., 2023), VisualWebArena (Koh et al., 2024) construct realistic and reproducible web environments to facilitate the development of web agents. There are also various studies to improve the web understanding or grounding capabilities of MLLMs (Gao et al., 2024; Kil et al., 2024) or develop agents for autonomous web navigation (Hong et al., 2023; Zheng et al., 2024; Cheng et al., 2024; He et al., 2024). Despite their success, the community still lacks a comprehensive evaluation of MLLMs' basic performance in web scenarios, including perception, understanding, grounding, and reasoning.

# 3 The `VisualWebBench` Benchmark

## 3.1 Overview of `VisualWebBench`

We present `VisualWebBench`: a multimodal benchmark designed to thoroughly evaluate the understanding and grounding capabilities of MLLMs in web scenarios. The proposed `VisualWebBench` possesses the following features: 1) **Comprehensiveness**: `VisualWebBench` spans 139 websites with 1.5K samples, encompassing 12 different domains (e.g., travel, sports, hobby, lifestyle, animals, science, etc.) and 87 sub-domains. 2) **Multi-granularity**: `VisualWebBench` assesses MLLMs at three levels: website-level, element-level, and action-level. 3) **Multi-tasks**: `VisualWebBench` encompasses seven tasks designed to evaluate the understanding, OCR, grounding, and reasoning capabilities of MLLMs. 4) **High quality**: Quality is ensured through careful human verification and curation efforts. The domain distribution and statistics of our benchmark are presented in Figure 3.

## 3.2 Website Selection

To ensure comprehensive coverage across diverse domains and top-ranking websites, our website selection process is conducted based on SimilarWeb[1]. We start from 12 top-level domains in SimilarWeb such as Science, Engineering, Sports, Lifestyle, and more, which are subsequently broken down into 87 sub-domains. Then we manually select representative websites from the top-5 most ranking websites in each sub-domain. Our selection criteria prioritize websites with rich interactive elements, including images and buttons, while excluding those that have been used in prior web agent benchmarks like Mind2Web and WebArena. We use Playwright[2] to render and save the websites automatically.

---

[1]https://www.similarweb.com
[2]https://github.com/microsoft/playwright

### 3.3 Task Construction

This section details the proposed seven tasks of `VisualWebBench` and the process of constructing data for each task; examples are shown in Figure 1.

**Meta Description Generation.** This task aims to evaluate the MLLMs' ability to comprehend and summarize the content of a webpage screenshot. The meta description, i.e., `<meta name="description">` tags in the head section of HTML, is a brief snippet of text that helps humans or search engines understand the content of websites. However, the quality of extracted meta descriptions cannot be ensured and their styles are pretty different on diverse websites. For example, some meta descriptions only consist of a list of keywords or a short title of the website, instead of a natural language description. Hence, we instruct GPT-4V to generate a better meta description, given both the screenshot and the extracted meta description. The final descriptions are verified and curated by the authors.

**WebQA.** To assess the understanding capabilities of MLLMs in the web scenario, `VisualWebBench` involves a webpage QA task, where the MLLM will answer open questions that demand a thorough comprehension of the visual layout. Human annotators are instructed to examine each screenshot and craft up to five challenging questions which satisfy: 1) A degree of reasoning ability is required to answer the question, 2) The answers should be precise and objective.

**Heading OCR.** This task requires MLLMs to locate and recognize the text of the heading of a website. Different from the traditional OCR task where a target element is given, as shown in Figure 1, the input of heading OCR is simply a raw screenshot, and the expected output is the heading content. The ground-truth target is automatically extracted from the first `<h1>` element in the HTML.

**Element OCR.** This task evaluates the capability of MLLMs to conduct OCR on lengthy texts. Firstly, we traverse the HTML DOM tree and extract the bounding boxes and textual description of each element Then, we select elements whose text descriptions comprise over 20 words. The task input consists of a screenshot with a bounding box indicating the position of the element to be recognized.

**Element Grounding.** Grounding, or Referring Expression Comprehension (REC), is a crucial image-text alignment capability, particularly for MLLMs interacting with web environments. Given a description of an HTML element, MLLM needs to locate the corresponding region in the screenshot. However, our preliminary studies reveal that current MLLMs struggle to directly give the coordinate of the target's bounding box (see 4.6). Inspired by Yang et al. (2023), we adopt a simplified setting where eight candidate bounding boxes are presented. Differently, the candidate elements here are extracted automatically using Playwright, with each assigned an alphanumeric ID. MLLMs are then prompted to select the box that best matches the given element description. The element description, golden bounding box, and negative bounding boxes of randomly chosen elements are automatically extracted from the webpage.

**Action Prediction.** This task asks MLLMs to predict the title of the redirected website after clicking an element, in a multi-choice QA way. In terms of the construction process, firstly, we employ Playwright to click all clickable elements within the web page and save the `<title>` or `<meta name="title">` tag as titles of new redirected web pages. Subsequently, we randomly sample seven additional elements distinct from the target element and take the titles of their respective redirect destinations as negative choices. Cases where a click does not lead to a title change are omitted from consideration. The task presents input in the form of screenshots highlighting the clickable target with a red bounding box. Accompanying each screenshot is eight choices, each labeled with a letter. The ground truth output is the letter corresponding to the correct answer.

**Action Grounding.** In addition to directly grounding elements from their descriptions, we further introduce the action grounding task. In this task, the MLLMs are given a human instruction, such as "search for the hotels in NYC", and are prompted to determine the correct element to click to fulfill the instruction. Similar with **Element Grounding**, MLLMs take in a screenshot containing bounding boxes of eight candidate elements and select the

| Model | Website | | | Element | | Action | | Average |
|---|---|---|---|---|---|---|---|---|
| | MetaGen | WebQA | HeadOCR | OCR | Ground | Prediction | Ground | |
| General MLLMs | | | | | | | | |
| Otter | 5.3 | 0.7 | 3.5 | 0.5 | 0.7 | 14.6 | 0.0 | 3.6 |
| InstructBLIP-13B | 11.6 | 5.2 | 7.6 | 6.0 | 11.4 | 11.4 | 17.5 | 10.1 |
| BLIP-2 | 11.0 | 5.2 | 20.6 | 2.6 | 15.5 | 14.9 | 8.7 | 11.2 |
| Fuyu-8B | 3.5 | 5.2 | 5.8 | 12.4 | 19.4 | 13.2 | 15.5 | 10.7 |
| Yi-VL-6B | 8.0 | 14.3 | 43.8 | 3.5 | 16.2 | 13.9 | 13.6 | 16.2 |
| LLaVA-1.5-7B | 15.3 | 13.2 | 41.0 | 5.7 | 12.1 | 17.8 | 13.6 | 17.0 |
| mPLUG-Owl2 | 12.7 | 19.9 | 51.6 | 7.2 | 11.9 | 23.1 | 3.9 | 18.6 |
| LLaVA-1.5-13B | 20.0 | 16.2 | 41.1 | 11.8 | 15.0 | 22.8 | 8.7 | 19.4 |
| SPHINX | 13.7 | 11.6 | 48.1 | 7.7 | 18.4 | 14.2 | 22.3 | 19.4 |
| Qwen-VL | 21.8 | 32.2 | 48.4 | 13.4 | 14.0 | 26.7 | 10.7 | 23.9 |
| CogVLM | 16.6 | 30.6 | 65.9 | 10.0 | 17.7 | 11.7 | 23.3 | 25.1 |
| VILA-13B | 12.7 | 28.8 | 67.9 | 12.6 | 16.5 | 36.3 | 16.5 | 27.3 |
| DeepSeek-VL-7B | 18.1 | 30.0 | 63.4 | 18.1 | 16.2 | 35.2 | 15.5 | 28.1 |
| LLaVA-1.6-7B | 27.0 | 39.8 | 57.3 | 54.8 | 31.7 | 30.6 | 10.7 | 36.0 |
| LLaVA-1.6-13B | 26.5 | 44.5 | 52.8 | 56.1 | 31.7 | 48.4 | 15.5 | 39.4 |
| LLaVA-1.6-34B | 24.3 | 48.2 | 67.1 | 71.9 | 43.1 | 74.0 | 25.2 | 50.5 |
| Gemini 1.0 Pro | 25.0 | 55.5 | **75.1** | 65.4 | 44.3 | 26.7 | 43.7 | 48.0 |
| Gemini 1.5 Pro | 31.6 | 69.0 | 54.5 | 76.6 | **70.0** | **74.4** | **77.7** | 64.8 |
| Claude Sonnet | 28.9 | **81.8** | 70.3 | **89.2** | 68.8 | 63.4 | 58.3 | **65.8** |
| Claude Opus | 26.7 | 75.4 | 63.7 | 87.1 | 57.7 | 60.4 | 38.8 | 58.5 |
| GPT-4V(ision) | **34.5** | 75.0 | 68.8 | 62.8 | 67.5 | 67.6 | 75.7 | 64.6 |
| GUI Agent MLLMs | | | | | | | | |
| SeeClick | 0.0 | 19.6 | 34.8 | 0.0 | 9.9 | 1.8 | 1.9 | 9.7 |
| CogAgent-Chat | 16.3 | 53.3 | 20.2 | 32.4 | 41.6 | 13.5 | 23.3 | 28.7 |

Table 1: Overall results of different models on `VisualWebBench` benchmark. The best-performing model is **in-bold**, and the second best is underlined. The maximum of the metrics is 100.

most appropriate one. The task data is completed by seven experienced annotators, and an annotation tool is developed to streamline the annotation workflow. Further details about the annotation tool and the annotation process can be found in Appendix C.

All tasks above adopt a VQA-style formulation similar to customary multimodal benchmarks. All screenshots in `VisualWebBench` are unified in a standard 1280-pixel width. All samples of our benchmark undergo careful verification and curation through a collaborative effort and a division of tasks by two authors. See Appendix B for more details.

### 3.4 Evaluation Metrics

We adopt different evaluation metrics for different tasks in `VisualWebBench`. For open-ended generation tasks, ROUGE-L (Lin, 2004) is used to measure the quality of the generated responses. For the WebQA task, SQuAD style F1 (Rajpurkar et al., 2016) is employed as the evaluation metric. For multiple-choice tasks, we measure accuracy.

## 4 Experiments

### 4.1 Evaluated MLLMs

We evaluate 16 open-source general MLLMs on `VisualWebBench` (See Appendix D for model details). By default, for each model family, we use the largest available checkpoint. We consider three scales of LLaVA, 7B, 13B, and 34B, for model scaling analysis. Several strong close-source MLLMs, Gemini Pro (Google et al., 2023), Claude series (Anthropic, 2024), and GPT-4V(ision) (OpenAI, 2023), are also included for evaluation.

Recent studies have introduced several MLLMs tailored to create agents for GUI tasks, such as web and smartphones (Cheng et al., 2024; Hong et al., 2023; Gao et al., 2024). Therefore, we consider two open-source GUI-specialized MLLMs for evaluation: SeeClick (Cheng

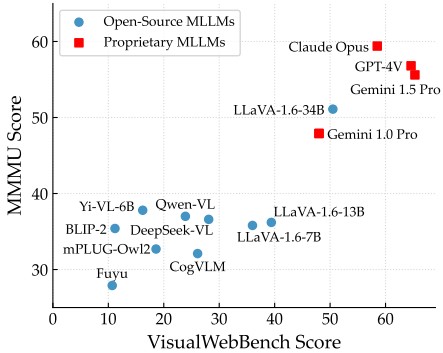
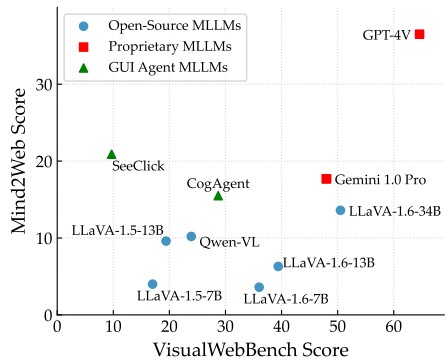

Figure 4: Scores of MLLMs on `VisualWebBench` and MMMU.

Figure 5: Scores of MLLMs on `VisualWebBench` and Mind2Web.

et al., 2024) is developed by GUI grounding pretraining based on Qwen-VL (Bai et al., 2023). CogAgent (Hong et al., 2023) is built upon CogVLM (Wang et al., 2023), focusing on GUI interpretation and navigation, with support for high-resolution image inputs.

## 4.2 Main Results

In this section, we present a comprehensive comparison of different MLLMs on `VisualWebBench` in Table 1. From the results, we highlight the following findings.

**Challenging Nature of Web Tasks:** Even the powerful GPT-4V achieves an average score of only 64.6 on `VisualWebBench`, leaving ample room for improvement. For tasks requiring strong reasoning and grounding abilities (Action Prediction and Action Grounding), many MLLMs struggle to surpass random chance (12.5). This underscores that current models cannot effectively handle many tasks within the web scenario.

**Disparity between Open-source and Proprietary MLLMs:** GPT-4V, Gemini 1.5 Pro and Claude outperform open-source MLLMs including GUI agent MLLMs by a large margin, highlighting a discernible gap in the capabilities of current open-source MLLMs compared to proprietary ones like GPT-4V. Meanwhile, LLaVA-1.6-34B achieves a commendable result (50.5) and beats all other open-source MLLMs, even outperforming the performance of Gemini 1.0 Pro (48.0). Notably, we find Claude Sonnet surpasses Opus on all tasks in `VisualWebBench`, suggesting that Sonnet may possess more powerful capabilities in web scenarios.

**Scaling Leads to Better Performance:** Compared with the 7B and 13B versions of LLaVA-1.6, the 34B model achieves a performance boost across almost all tasks, reaching an average score of 50.5. Although there are factors other than scale, such as different backbone LLMs, this indicates that increasing model size is a promising avenue for enhancing the capabilities of open-source MLLMs in web-related tasks.

**General MLLMs vs. GUI Agent MLLMs:** SeeClick and CogAgent are two MLLMs pre-trained on GUI grounding tasks. However, we observe that these GUI agent MLLMs do not exhibit significant performance improvement. For example, SeeClick fails to outperform Qwen-VL, its base MLLM, across all tasks. Notably, we find these models suffer catastrophic forgetting (Wang et al., 2024) on general instruction following capability after training on GUI grounding data. These results underscore the necessity for more general GUI-specific training techniques to enhance the MLLMs' performance in the web scenario. To further investigate the effectiveness of GUI grounding training, we perform a comprehensive comparison of various grounding settings in Section 4.6.

### 4.3 Correlations with General Scenario and Agent Benchmarks

We delve into the relationship between the performance of MLLMs in the web scenario and that in general and agent scenarios. Specifically, we use MMMU (Yue et al., 2023) as the proxy of MLLMs' capability in general scenario[3], and Mind2Web (Deng et al., 2024) is used for the evaluation of the agent scenario.

While Figure 4 somewhat suggests some correlation between VisualWebBench score and MMMU score, the relationship does not appear to be significant. In other words, performing well in the general domain does not necessarily guarantee the same trend in the web scenario. For example, while Yi-VL-6B and BLIP2 outperform CogVLM on MMMU, they fall short in achieving a good score on VisualWebBench. It is also noteworthy that LLaVA-1.6-34B performs well on both tasks, nearly matching the performance level of GPT-4V.

As illustrated in Figure 5, generally, VisualWebBench scores are higher than those of Mind2Web[4], which demonstrates that there still exists large room for improvement of agent ability empowered by webpage understanding, grounding abilities, as well as other abilities like planning. GUI agent MLLMs tend to exhibit overfitting in terms of agent capability, resulting in underperformance in understanding web pages.

Moreover, in Figure 6, we conduct an in-depth correlation analysis between MMMU and VisualWebBench seven subtasks, as well as a similar analysis between Mind2Web and VisualWebBench tasks. From the results in Table 1, we have seven sets of metrics (columns), and Pearson correlation coefficients are calculated between every two columns. For MMMU, the correlations are generally strong. Specifically, the two subtasks requiring heavy reasoning, WebQA and Action Prediction, strongly correlate with MMMU. For Mind2Web, the correlation between scores on

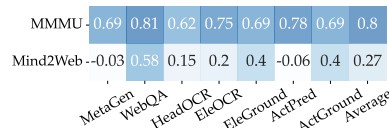

Figure 6: Correlations between VisualWebBench and MMMU, Mind2Web, respectively.

VisualWebBench and those on Mind2Web is low, even exhibiting two negative correlations in Meta Description Generation and Action Grounding. These findings suggest that VisualWebBench offers a different evaluation perspective for MLLMs in the web scenario.

### 4.4 Correlation Between VisualWebBench Tasks

Figure 7 illustrates the correlations between tasks in VisualWebBench. This analysis reveals a strong correlation among specific tasks, namely Meta Description Generation, WebQA, and Element OCR, all demanding a comprehensive understanding of textual content within webpages. In contrast, Action Prediction and Action Grounding tasks exhibit a minimal correlation, implying distinct skill sets necessary for predicting action outcomes versus pinpointing elements for actions. Moreover, Action Grounding seems to be less correlated with all other non-grounding tasks, highlighting its distinctive and specialized skill requirements.

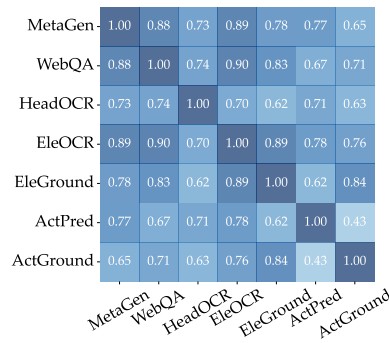

Figure 7: Correlations between 7 subtasks in VisualWebBench.

### 4.5 Analysis of Image Resolution

Most current MLLMs can only process low-resolution images, typically 448×448. However, the screenshots in VisualWebBench are captured in high resolution (1280 pixels in width), presenting challenges in identifying intricate details at lower reso-

---

[3]The overall score on the validation set of MMMU is used for comparison.
[4]Detailed experimental results are included in Appendix E.

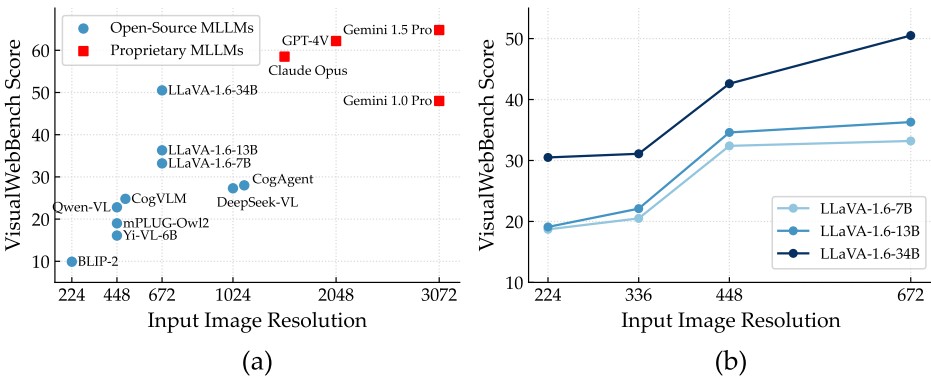

Figure 8: The effect of image resolution on VisualWebBench score.

lutions. In this section, we explore the effects of input resolution on model performance. We plot the relation of max input image resolution and `VisualWebBench` scores for different MLLMs in Figure 8(a). Notably, MLLMs with higher input resolution generally achieve higher scores. For instance, DeepSeek-VL with 1024×1024 resolution achieves a higher score than Qwen-VL with 448×448 resolution.

Based on LLaVA-1.6 series models, we further conduct a formal ablation study on input image resolution. As depicted in Figure 8(b), a significant performance improvement is observed as input image resolutions increase for all three model sizes. Additionally, the models exhibit greater benefits when increasing resolution from 336 to 448, compared with from 448 to 672. This finding suggests that, for LLaVA-1.6, a resolution of 448×448 stands as the minimal requirement to achieve adequate performance in web-related tasks.

## 4.6 Analysis of Grounding Capability

In our experiments in Section 4.2, for Element and Action Grounding tasks, we provide eight candidate elements and use a multiple-choice setting to evaluate different MLLMs. However, in many applications, the screenshots of webpages cannot be annotated with candidate bounding boxes. Hence, we evaluate the grounding capability in unannotated images by framing the grounding tasks as a Referring Expression Comprehension (REC) problem, where the MLLMs must generate the position (bounding box $[x_1, y_1, x_2, y_2]$ or central point coordinate $[x, y]$) of the selected HTML element. For the setting of the bounding box, we follow the standard REC task and use $AP_{50}$ (Lin et al., 2014) as the metric. For the setting of point prediction, a predicted point is regarded as correct if it falls into the true bounding box.

As Table 2 shows, GUI agent MLLMs significantly outperform general MLLMs (e.g., LLaVA-1.6 and GPT-4V) in generating the positions (Bbox or point) of target elements, confirming the efficacy of grounding pre-training through the point or bounding box prediction. For other MLLMs that have been trained on general grounding data like RefCOCO, they still fail to accurately give the coordinates of the correct elements.

## 4.7 Case Studies

We show a few case studies for LLaVA-1.6-34B, CogAgent, and GPT-4V on action prediction and action grounding tasks. For the action prediction task (Figure 9), CogAgent generates a wrong choice without any explanation, while LLaVA locates a wrong element ("Go" button). Notably, GPT-4V shows a reasonable thinking process and the correct answer. For action grounding (Figure 10), despite LLaVA and GPT-4V generating reasonable thought processes, all three models fail to answer correctly. See Appendix F for more case studies.

| Model | Element Ground | | | Action Ground | | |
| --- | --- | --- | --- | --- | --- | --- |
| | Multi-choice | Bbox | Point | Multi-choice | Bbox | Point |
| Fuyu-8B | 19.4 | 0.0 | 0 | 15.5 | 0.0 | 0.0 |
| VILA-13B | 16.5 | 1.0 | 7.8 | 16.5 | 0.0 | 5.9 |
| LLaVA-1.6-7B | 31.7 | 0.2 | 4.6 | 10.7 | 0.0 | 5.9 |
| LLaVA-1.6-13B | 31.7 | 0.0 | 0.7 | 15.5 | 1.0 | 5.9 |
| LLaVA-1.6-34B | 43.1 | 1.7 | 10.7 | 25.2 | 3.0 | 10.9 |
| Qwen-VL | 14.0 | 1.5 | 3.9 | 10.7 | 0.0 | 3.0 |
| GPT-4V(ison) | 67.5 | 0.2 | 1.5 | 75.7 | 0.0 | 1.0 |
| SeeClick | 9.9 | 0.0 | 70.0 | 1.9 | 0.0 | 42.6 |
| CogAgent-Chat | 41.6 | 29.3 | 46.3 | 23.3 | 36.6 | 58.4 |

Table 2: Three evaluation settings for grounding tasks. "Multi-Choice" is the default setting in `VisualWebBench`. "Bbox" and "Point" denote the setting of predicting the coordinate of the target bounding box and central point, respectively.

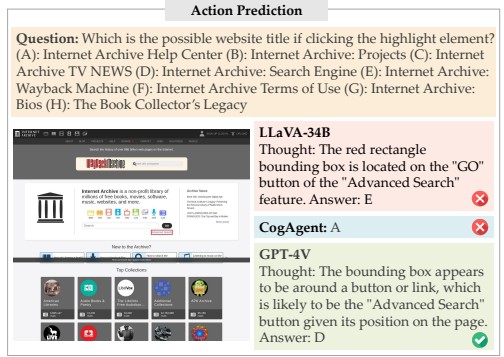

Figure 9: Case study of Action Prediction.

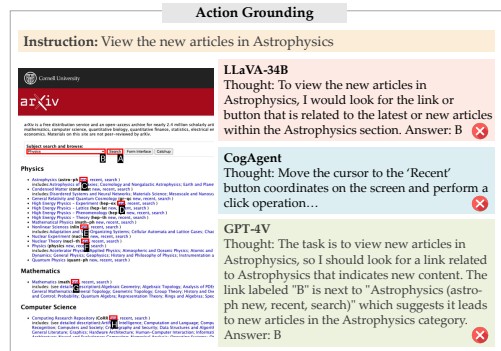

Figure 10: Case study of Action Grounding.

# 5 Conclusion

In this work, we introduce `VisualWebBench`: a comprehensive benchmark to evaluate the web page understanding and grounding capabilities of MLLMs. `VisualWebBench` encompasses seven tasks spanning three different levels covering web page, element, and user action. Unlike existing benchmarks, our benchmark aims to comprehensively evaluate MLLMs in web contexts, including understanding, OCR, grounding, and reasoning. Our evaluation of 14 open-source MLLMs, Gemini Pro, Claude Sonnet, Claude Opus, and GPT-4V(ision) shows the substantial challenges posed by realistic web tasks. Further analysis highlights several limitations of current MLLMs, including inadequate grounding in text-rich environments and subpar performance with low-resolution image inputs. We believe `VisualWebBench` will serve as a catalyst for further exploration in the development of MLLMs towards artificial general intelligence.

# Acknowledgement

The authors would thank Boyuan Zheng, Shuyan Zhou, Yizhong Wang, and Jie Huang for their insightful discussions and comments. The authors would also thank seven annotators for their help in annotating the action grounding task samples. Xiang Yue is supported by Carnegie Bosch Fellowship. Junpeng Liu and Wai Lam are supported by a grant from the Research Grant Council of the Hong Kong Special Administrative Region, China (Project Code: 14200719).

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

## A Data Construction and Annotation Details

For the WebQA task, two co-authors carefully examined each screenshot and formulated questions and corresponding answers. For the Action Grounding task, the annotation process was carried out by seven experienced researchers, including four co-authors. The construction processes for the other subtasks were automated through program execution and were relatively straightforward.

## B Data Verification and Curation

All samples of our benchmark undergo careful verification and curation through a collaborative effort and a division of tasks by two authors. The process encompasses:

- Ensuring the main content within screenshots remains unobscured by advertisements or intrusive banners.
- Verifying that the meta descriptions describe the most important information generally of the websites.
- Headings of websites are correctly extracted for Heading OCR.
- The annotated bounding boxes properly encapsulate the target web element description for Element OCR and Element Grounding.
- The annotated bounding boxes are well aligned with the title of redirected websites for Action Prediction.
- The instructions are appropriately matched with their annotated bounding boxes for Action Grounding.

## C Annotation Tool of Action Grounding

We developed an annotation tool to facilitate the annotation of the action grounding task. The annotation procedure is as follows:

1. Learn about what the shown website is for, based on the presented website descriptions, and you may still need to search for the website name in Google to have a better understanding.
2. Refer to action description examples generated by GPT-4V, and then write your instruction. Then, click "Confirm instruction". Please make your instructions diverse, and do not write too many instructions like "search for an item".
3. Move the Mouse to hover over the corresponding element that will be interacted with to accomplish the action description, then press key "s" (instead of CLICK) to select it. After that, a green rectangle will be shown to indicate the selected element. Note that the element should be interactive (e.g., clickable or inputtable, etc.).
4. Confirm that the selected element (indicated by a blinking green rectangle) correctly corresponds to the action description and click the "submit" button, then click "allow" to allow screen capture.

## D Details of Evaluated MLLMs

We consider various general large multimodal models. By default, for each model family, we use the latest, largest, and best-performing available checkpoint to date. *(i)* BLIP-2 (Li et al., 2023c) series bridges the vision-language modality gap with a lightweight Q-Former. *(ii)* InstructBLIP (Dai et al., 2024) further performs vision-language instruction tuning based BLIP-2 models. *(iii)* mPLUG-Owl2 (Ye et al., 2023) adapts a modularized network to facilitate modality collaboration while preserving specific features. *(iv)* Otter (Li et al., 2023a) has improved instruction following and in-context learning capabilities. *(v)* VILA (Lin et al., 2023a) is pretrained with interleaved image-text data at scale. *(vi)* Fuyu (Bavishi et al., 2023) is a decoder-only transformer and treats image tokens like text tokens. *(vii)* SPHINX (Lin

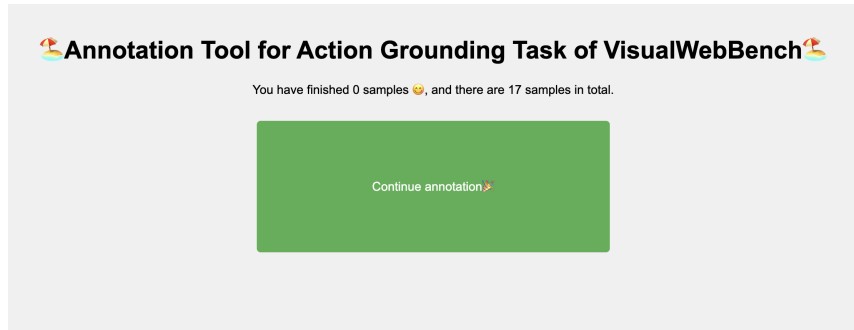

Figure 11: Illustration of the annotation tool (1).

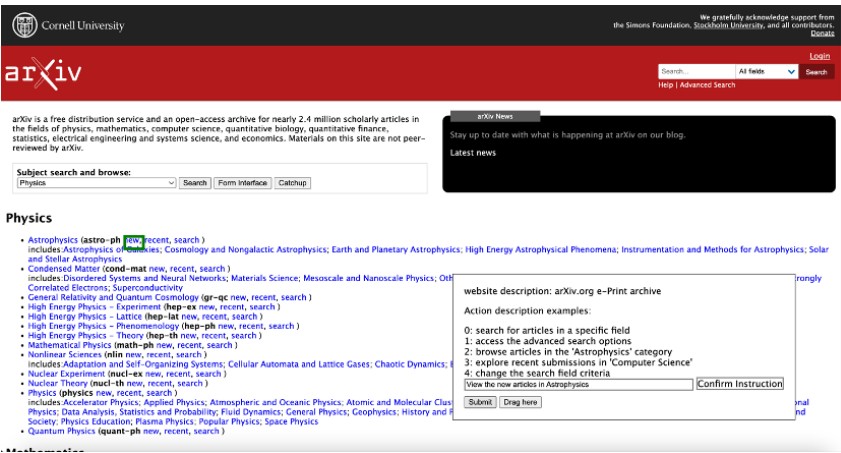

Figure 12: Illustration of the annotation tool (2).

et al., 2023b) mixes different tuning tasks, and visual embeddings to build a versatile MLLM. *(viii)* LLaVA-1.5 (Liu et al., 2023a) combines a vision encoder and Vicuna for general-purpose visual and language understanding, and LLaVA-1.6 (Liu et al., 2024) family is the enhanced version with improved image resolution, reasoning, OCR, and world knowledge. We consider three scales: Vicuna-7B, Vicuna-13B, and Hermes-Yi-34B for model scaling analysis. *(ix)* Qwen-VL (Bai et al., 2023) introduces trainable query embeddings and single-layer cross-attention module to bridge the modalities. *(x)* DeepSeek-VL (Lu et al., 2024) incorporates a hybrid vision encoder to processe high-resolution images. *(xi)* Yi-VL (Young et al., 2024) connects the vision encoder with MLLM with a simple MLP projection module and undergoes a three-stage training process. *(xii)* CogVLM (Wang et al., 2023) bridges the modality gap by a trainable visual expert module in the attention and FFN layers of the transformer. We also include Gemini Pro (Google et al., 2023), Claude Sonnet, Claude Opus (Anthropic, 2024), and GPT-4V(ision) (OpenAI, 2023) for comparison.

For all MLLMs, we set the temperature to 0.0 for deterministic generations. All experiments are conducted on NVIDIA A100 80G GPUs.

## E  `VisualWebBench` vs. **Mind2Web**

Table 3 details the scores of MLLMs on `VisualWebBench` and Mind2Web.

## F  **Additional Case Studies**

In Figure 13, CogAgent cannot understand the instruction correctly and output unrelated content. LLaVA-34b captures the instruction, yet generates a wrong answer. In contrast,

|              | VisualWebBench | Mind2Web |
|--------------|:--------------:|:--------:|
| SeeClick     | 9.7            | 20.9*    |
| Qwen-VL      | 23.9           | 10.2*    |
| CogAgent     | 28.7           | 15.5†    |
| LLaVA-1.5-7B | 17.0           | 4.0      |
| LLaVA-1.5-13B| 19.4           | 9.6      |
| LLaVA-1.6-7B | 36.0           | 3.6      |
| LLaVA-1.6-13B| 39.4           | 6.3      |
| LLaVA-1.6-34B| 50.5           | 13.6     |
| Gemini 1.0 Pro | 48.0         | 17.7†    |
| GPT-4V(ison) | 64.6           | 36.5†    |

Table 3: The comparison between scores of VisualWebBench and Mind2Web. * indicates the results taken from Cheng et al. (2024), while † denotes those taken from Zheng et al. (2024).

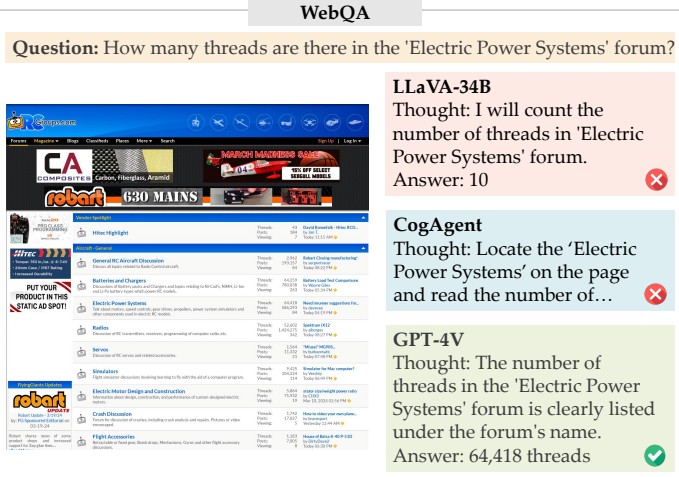

Figure 13: Case study of WebQA.

both the thinking process and the answer of GPT-4V are correct. This demonstrates the strong understanding and reasoning ability of GPT-4V.

Case studies for Meta Description Generation, Heading OCR, Element Grounding, and Element OCR are shown in Figure 14, Figure 15, Figure 16, and Figure 17.

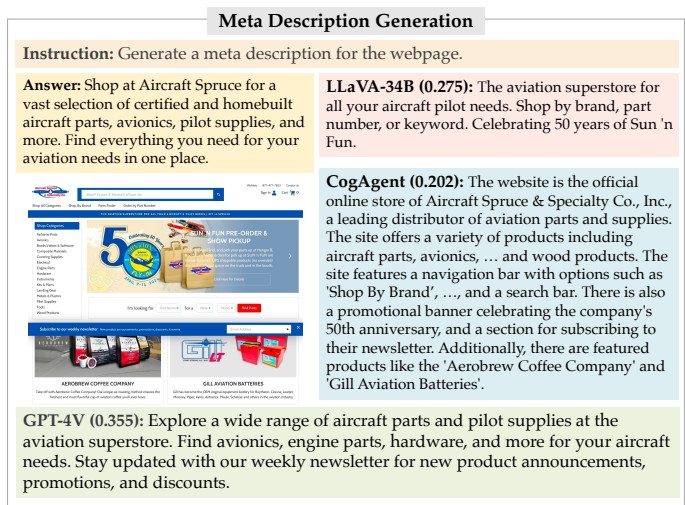

Figure 14: Case study of Meta Description Generation.

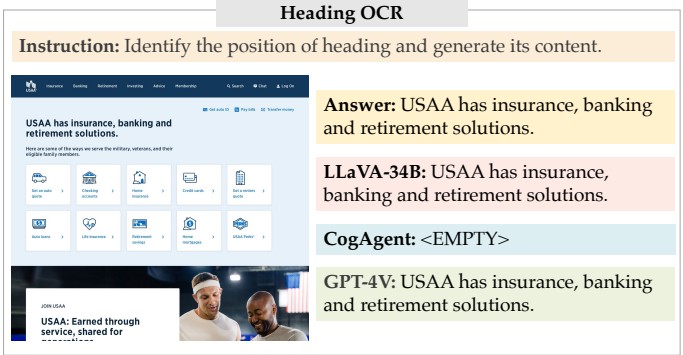

Figure 15: Case study of Heading OCR.

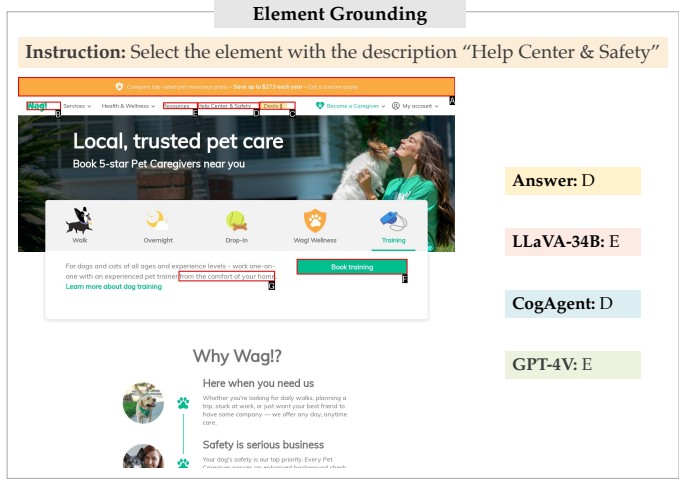

Figure 16: Case study of Element Grounding.

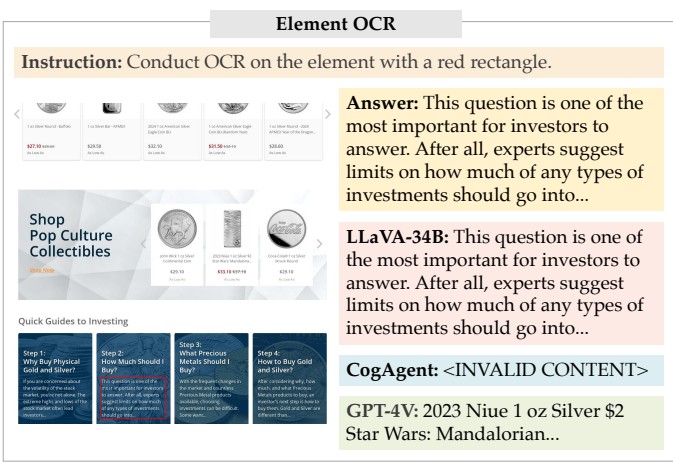

Figure 17: Case study of Element OCR.

