# OpenReview forum: "VisualWebBench: How Far Have Multimodal LLMs Evolved in Web Page Understanding and Grounding?"
_colmweb.org/COLM/2024/Conference — COLM_

### Official Review · Reviewer_VZGD · 2024-05-06

**Rating:** 7
**Confidence:** 4
**Ethics Flag:** 1

**Summary:**

The paper proposes a new evaluation dataset called WebBench that tries to cover the issues of existing benchmarks being: failing to capture the unique features of web pages and to measure fine grained capabilities such as OCR, understanding and grounding.

WebBench consists of 1.5k instances covering seven tasks: captioning, webpage QA, heading OCR, element OCR, element grounding, action prediction and action grounding.

The authors evaluate 14 open-source MLLMs covering closed and open source models.

The benchmark is constructed using different processes depending on the task. Some use LLMs and human verification, others are completely carried out by human annotators.

**Questions To Authors:**

Is it correct saying that the number of unique screenshots is 139?

Why is a discussion / related work on methods that relies on the DOM/source code to carry out tasks on web pages not in the paper?
E.g. DocLLM: A layout-aware generative language model for multimodal document understanding

For the heading OCR task, how often the H1 header is the right one?

Why is Gemini Pro missing from Figure 4 and 6?

What about using model-based metrics or critics for evaluating the open ended generation tasks?

**Reasons To Accept:**

The benchmark covers a sufficiently wide range of tasks on website screenshots and it is definitely a useful resource for evaluating multimodal language models.

The evaluation includes the main closed and open source LLMs.

**Reasons To Reject:**

There is one issue in the way the evaluation data is produced for one task, namely captioning. In this task, metadata that is mostly a sequence of keywords is converted into a more natural description with GPT-4V. In addition, the latter is seeing both screenshot and metadata. At evaluation time, each model is asked to generate the description. This whole process gives an unfair advantage to GPT-4V, which is exacerbated by using ROUGE-L as the evaluation metric for open-ended tasks. It is fair to say that gold and predicted descriptions would exhibit similar structure and wording. This raises a question about how much GPT-4V relies on the keywords to output the description, and how much information is actually coming from the screenshot. Maybe the model relies more on the image?

On a similar note, the appendix shows that action descriptions are also suggested to annotators by GPT-4V. What if the model is generating and biasing the annotators towards actions that it can successfully perform? Would this possibility conflate GPT-4V performance on this task too?

---

> ### Author Rebuttal · Authors · 2024-05-31
>
> We appreciate that the reviewer recognizes our benchmark as a valuable resource for evaluating multimodal language models in webpage understanding!
>
> **Bias Introduced by GPT-4V**: We considered the potential bias introduced by GPT-4V during our data annotation process. **Consequently, we have undertaken further data verification by human annotators as detailed in Appendix B.** For the action grounding task, as described in Appendix A, annotators were encouraged to write their own instructions. Ultimately, **80% of annotated action descriptions are different** from the ones initially generated by GPT-4V.
>
> **Unique Screenshot Numbers:** The total number of unique screenshots is 332, instead of 139. While captioning, heading OCR, and action grounding tasks generate one sample per website, other subtasks construct multiple samples for each website.
>
> **Related Work:** We discussed the related work that relies on the DOM/source code to carry out tasks on web pages **in Introduction and Section 2.2** (e.g., web agent benchmarks like MiniWob++, WebShop, Mind2Web, and WebArena). **The mentioned work DocLLM, focuses on documents and tables understanding rather than web pages, which clearly has different goals from our work.** We will include the comparison between DocLLM and our WebBench in the revised version.
>
> **Automatically Extracted h1 Tag**: As indicated in Figure 3, **only 46 out of 139 samples** are properly displayed in screenshots and can be considered good headings for web pages. **This underscores the necessity of our human verification process.**
>
> **Results of Gemini Pro in Figures 4 and 6**: We have uploaded the updated figures to the anonymous link. (https://postimg.cc/gallery/F0yRZ0f).
>
> **Model-Based Metrics for Open-Ended Generation Tasks**: Given that model-based metrics may introduce **uninterpretable bias and additional computational cost**, we prefer lightweight and rule-based evaluation metrics. For instance, for the WebQA task, annotators are instructed to propose objective questions with short answers (less than five words in most cases), making SQUAD-F1 the appropriate metric.

---

> > ### Comment · Reviewer_VZGD · 2024-06-05
> >
> > Thank you for your replies and for updating the figures. I think the current score captures my excitement about the paper.

---

> > > ### Author Response · Authors · 2024-06-05
> > > **Thank you!**
> > >
> > > Thank you so much for your positive feedback! We sincerely express our gratitude for your constructive comments and suggestions!
> > >
> > > Best,
> > > Authors

---

### Official Review · Reviewer_3CW7 · 2024-05-11

**Rating:** 8
**Confidence:** 4
**Ethics Flag:** 1

**Summary:**

The paper proposes WebBench, a fine-grained benchmark for webpage understanding. Specifically, the paper argues that existing web benchmarks for multimodal LLMs are focused on end-to-end or agent capabilities, conflating multiple challenges in one task and thus hard to pin down what is truly challenging for a tested model.

WebBench has 1.5k samples from 39 websites.  It breaks down web understanding into 3 levels: (1) website (2) element and (3) action. Under each level, there are elementary skills needed such as HeadOCR and Captioning for (1) and grounding for (2).

The authors evaluated both closed-source and open-source multimodal LLMs on WebBench and found even the SOTA model (GPT-4V) achieves only ~65%, indicating room for improvement.

**Reasons To Accept:**

+ Overall I find WebBench as a worthwhile effort to break down challenges in web understanding. Web understanding is known to be complex and often requires understanding of different levels and sometimes agentic workflow. It makes a lot of sense to test these capabilities individually instead of having one end-to-end task which can end up being too challenging and uninterpretable.

+ The authors tested a wide range of models, especially open-sourced ones. One nit-picking suggestion is to include more state-of-the-art closed source models such as Gemini 1.5 Pro and Claude 3 Opus.

**Reasons To Reject:**

I do not have a major reason to reject.

---

> ### Author Rebuttal · Authors · 2024-05-31
>
> We appreciate the reviewer’s recognition of our efforts to break down the challenges in web understanding across a wide range of models. Below, we add the results of Gemini 1.5 Pro and Claude 3 Opus. **Overall, Gemini 1.5 Pro performs the best on the majority of the subtasks and achieves the highest average score.** Claude Opus shows exceptional performance on Element OCR. We will include the results and the analysis in our revision!
>
>
> | Model          | Caption  | WebQA    | HeadOCR  | EleOCR   | EleGround | ActPred  | ActGround | Average  |
> | -------------- | -------- | -------- | -------- | -------- | --------- | -------- | --------- | -------- |
> | Gemini 1.0 Pro | 25       | 55.5     | **75.1** | 65.4     | 44.3      | 26.7     | 43.7      | 48       |
> | Gemini 1.5 Pro | 31.6     | 69       | 54.5     | 76.6     | **70**    | **74.4** | **77.7**  | **64.8** |
> | Claude Opus    | 26.7     | **75.4** | 63.7     | **87.1** | 57.7      | 60.4     | 38.8      | 58.5     |
> | GPT-4V(ision)  | **34.5** | 75       | 68.8     | 62.8     | 67.5      | 67.6     | 75.7      | 64.6     |

---

### Official Review · Reviewer_RE1U · 2024-05-12

**Rating:** 7
**Confidence:** 5
**Ethics Flag:** 1

**Summary:**

This work presents WebBench, a new webpage understanding benchmark of just over 1.5k samples. The benchmark is designed to target seven different web page related tasks: captioning, webQA, heading OCR, element OCR, element grounding, action prediction and action grounding. Using the new dataset, 18 different model variants (of 16 unique model architectures) are evaluated. Along with the quantitative task evals, there is substantial analysis, including correlation analysis among the tasks themselves, as well as the correlation between other vision-language scores with WebBench aggregate scores and the relationship between WebBench scores and image resolution size. The work also includes a couple of qualitative examples of failure cases and ablation studies for grounding.

This paper presents high quality work with an appropriate amount of analysis and discussion. Its thorough benchmarking across many model variants is also appreciated. While the benchmark is not very large, this topic is a growing research area and having additional evaluation resources would positively impact the community.

**Questions To Authors:**

Questions
1. How is the correlation between tasks in Figure 5 calculated exactly?

Below are comments/suggestions for the authors
1. As mentioned in the reasons to reject, there are a couple of highly relevant related work that I would suggest adding to the prior work/citations:

[1] Yao et al. WebShop: Towards Scalable Real-World Web Interaction with Grounded Language Agents. NeurIPS 2022.

[2] Burns et al. A Suite of Generative Tasks for Multi-Level Multimodal Webpage Understanding. EMNLP 2023.

[3] Srinivasan et al. WIT: Wikipedia-Based Image Text Dataset for Multimodal Multilingual Machine Learning. ACM SIGIR 2021.

2. The last sentence in Figure 3 seems unnecessary, it was probably from an earlier version of the draft (I'd suggest removing).
3. Figure 2 is never referenced in text.

**Reasons To Accept:**

1. The paper proposes a well designed new benchmark for multimodal webpage understanding, consisting of a range of 7 different tasks. The tasks capture different levels of webpage understanding: the full webpage, element, and action level.
2. Along with the data there are substantial evals run across many models, providing comprehensive baselines, most of which are open source and easily reproduced.
3. There are numerous analyses and ablations provided, which verifying desirable qualities in the dataset and its utility in future work.

**Reasons To Reject:**

1. While more resources are certainly useful in this topic area, there are captioning and grounding datasets that would capture several of the tasks proposed in WebBench; either from prior human annotation efforts or automatically synthesized datasets. From the related work discussion and examples, it is not clear to me what distinguishes the dataset from prior work.

[1] Yao et al. WebShop: Towards Scalable Real-World Web Interaction with Grounded Language Agents. NeurIPS 2022.

[2] Burns et al. A Suite of Generative Tasks for Multi-Level Multimodal Webpage Understanding. EMNLP 2023.

[3] Srinivasan et al. WIT: Wikipedia-Based Image Text Dataset for Multimodal Multilingual Machine Learning. ACM SIGIR 2021.

2. Noise in task annotations. In the construction of the element grounding task, it's highly probable that there is noise in the annotations. For example, are there elements selected as negatives that could be congruent with the target element in action prediction task?

3. Another drawback is that it seems the tasks are annotated on separate samples, and ultimately there aren’t that many samples per task. This makes the benchmark a bit less well motivated because there are other datasets that cover the same task, and it would have been nice to have annotations for many tasks on a single screen, instead of them being non-overlapping / disjoint.

---

> ### Author Rebuttal · Authors · 2024-05-31
>
> We appreciate that the reviewer finds our benchmark well-designed, the experiments comprehensive, and the analyses insightful.
>
> **Differences from Existing Datasets:** 1) As described in the third paragraph of the introduction, web-agent benchmarks like Webshop [1] **focus on end-to-end capabilities without a fine-grained assessment of specific skills** such as OCR, semantic understanding, and grounding, which was also mentioned by **Reviewer f5BP (thanks!)**. 2) The WikiWeb2M dataset [2] includes only three captioning tasks on a webpage, and the WIT dataset [3] focuses on image-text retrieval. Additionally, **both datasets are limited to Wikipedia web pages and do not reflect the diverse scenarios web agents encounter**. In contrast, WebBench spans 139 most popular websites, covering 87 sub-domains and encompassing seven different sub-tasks, including captioning, OCR, web QA, and element or action grounding.
>
> **Noise in Task Annotation:** As detailed in Appendix B, **the human annotators undertook meticulous efforts to curate and verify all subtasks carefully**. For the element grounding subtask, we identified and removed the problematic cases, including those noted by the reviewer.
>
> **Not that many samples per task**: Our dataset contains multiple examples of different subtasks within a single screenshot. **We include more screenshots rather than annotating numerous examples within the same screenshot to ensure comprehensive coverage of diverse real-world scenarios.** Consequently, our dataset spans 139 of the most popular websites across 87 sub-domains. Notably, several widely adopted multimodal datasets prioritize diversity over quantity, such as MMVet with 218 samples and MME with 2374 samples \[4\]\[5\].
>
> [4] Yu, Weihao, et al. "Mm-vet: Evaluating large multimodal models for integrated capabilities." International Conference on Machine Learning, 2024, PMLR.
>
> [5] Fu, Chaoyou, et al. "MME: A Comprehensive Evaluation Benchmark for Multimodal Large Language Models." 2023, arXiv, eprint 2306.13394.
>
> **Calculation Detail of Correlations Between Subtasks in Figure 5:** From the results in Table 1, we have seven sets of metrics (columns), and the Pearson correlation coefficient is calculated between every two columns.
>
> **Typos**: We will correct the typos in the revised version.

---

### Official Review · Reviewer_f5BP · 2024-05-14

**Rating:** 8
**Confidence:** 4
**Ethics Flag:** 1

**Summary:**

This work presents “WebBench,” a new benchmark for evaluating multimodal language models (e.g., image, text → text), specifically focusing on applications involving reasoning over webpages. The benchmark consists of ~1.5K example prompts (webpage/input/response tuple) spanning 139 individual websites, and **7 task families** each with their own evaluation metric:

1. *Captioning* (ROUGE Score): Given a webpage screenshot, generate a natural language summary of the website (where ground-truth is a GPT-4V rewrite of the HTML “meta” tag)
2. *WebQA* (F1 Score): Given a question about the webpage (written by a human annotator) generate the answer (open-set generation).
3. *Heading OCR* (ROUGE Score): Generate the heading text for the given webpage (extracted from the first <h1> tag).
4. *Element OCR* (ROUGE Score): Given a webpage with a highlighted DOM element > 20 words (drawn bounding box), transcribe the given element in natural language.
5. *Element Grounding* (Multiple Choice Accuracy): Given a human-provided description of an HTML DOM element, and 8 potential candidates (drawn bounding boxes), select the candidate element that best matches the description.
6. *Action Prediction* (Multiple Choice Accuracy): Given a highlighted DOM element (with an href tag), select the title of the “target” webpage given a set of candidates (with negatives scraped from other elements of the same webpage).
7. *Action Grounding* (Multiple Choice Accuracy): Given a human-provided high-level instruction (e.g., “search for flights to Montreal”) and eight candidate DOM elements (drawn bounding boxes), identify the candidate element that is most relevant to the instruction (e.g., search bar for “Flights”).

Beyond curating this extensive benchmark, the work also evaluates and performs a thorough analysis of 14 state-of-the-art multimodal language models (14 open-source, 2 closed-source) across all tasks, including a stratified breakdown of how models behave as a function of scale (7B vs. 13B vs. 34B), input resolution, and manual error analysis. The appendix provides additional detail as to the data annotation and verification process. The full evaluation benchmark and evaluation harnesses for open-source models will be released.

**Questions To Authors:**

Were external annotators only used for the action grounding task? If not, could you provide a description of the annotation pipeline used for the other subtasks (similar to Appendix A)?

Beyond the webpage screenshots and input prompts provided in the benchmark, can you clarify what other metadata about the webpages you plan to release? Minimally, it would be great to get original URLs, access dates, as well as code for rendering websites (e.g., the Playwright pipeline described in the paper).

**Reasons To Accept:**

This is a strong paper introducing a new evaluation that 1) fulfills a demonstrated need given the recent proliferation of multimodal language models, and 2) clearly distinguishes itself from prior web-based evaluation benchmarks such as Web Shop and Visual WebArena by focusing on clearly defined and fine-grained “subtasks” necessary for reasoning over websites.

While the overall sample size per each of the 7 subtasks is on the smaller side, I do believe the existing examples are clean and high-signal. The effort the work makes to leverage human annotators to both provide input prompts and verify outputs is clear, and further strengthens the benchmark. Finally, the analysis of existing models demonstrates that this benchmark is not only valid, but that there are subtasks with a ton of “headroom” — showing that this evaluation will be useful to the field and those working on developing stronger MLLMs in the future.

**Reasons To Reject:**

There are a few minor issues that I believe the authors can easily fix for the final submission. First, it’s a bit strange to me that the evaluation metric for all of the OCR-based subtasks (e.g., Element OCR, Heading OCR) is ROUGE score rather than exact match (or F1 score); if we want high-fidelity OCR, shouldn’t we be penalizing our MLLMs for any generations that deviate from the ground-truth?

Furthermore, some of the details around the annotation procedure are missing (how annotators are selected, which annotators are used for generating which data per subtask, which data the authors of the work actually verified vs. external annotators). It would be really great to be specific and clear about these details in the final version of the paper.

---

> ### Author Rebuttal · Authors · 2024-05-31
>
> We appreciate the reviewer's recognition of our benchmark as high-quality, clearly defined, and well-motivated!
>
> **Metric for OCR Tasks:** We reported the F1 value of ROUGE-L because the average ground-truth length in our dataset is longer than that of common scene OCR datasets (e.g., Element OCR averages 39.5 words). Consequently, the longest common subsequence may be more appropriate than exact match for evaluation. For reference, we also provide exact match (EM), word-level F1, and ROUGE-L results below. **The EM metric is overly strict and does not accurately reflect OCR capabilities, while word-level F1 is similar to ROUGE-L.** For simplicity, we will continue to use ROUGE-L as the evaluation metric for the two OCR tasks, and we will include the following table in the appendix.
>
> |  | HeadOCR(ROUGE-L) | HeadOCR(EM) | HeadOCR(F1) | EleOCR(ROUGE-L) | EleOCR(EM) | EleOCR(F1) |
> |-|-|-|-|-|-|-|
> | LLaVA-1.6-7B | 57.3 | 39.1 | 57.3 | 54.8 | 4.1 | 55.4 |
> | LLaVA-1.6-13B | 52.8 | 43.5 | 52.8 | 56.1 | 8.6 | 56.6 |
> | LLaVA-1.6-34B | 67.1 | 47.8 | 67.1 | 71.9 | 13.9 | 72.2 |
> | Gemini 1.0 Pro|75.1|58.7|75.1|65.4|18|60.3|
> | Gemini 1.5 Pro|54.5|26.1|54.8|76.6|23.7|73.2|
> | Claude Opus|63.7|45.7|63.7|87.1|45.7|87.1|
> | GPT-4V(ision)|68.8|50|68.8|62.8|27.8|63|
>
> **Annotator Details:** For the WebQA task, **two co-authors** carefully examined each screenshot and formulated questions along with corresponding answers. For the Action Grounding task, the annotation process was carried out by **seven experienced researchers, including four co-authors**. We have included the annotator IDs for each sample in the anonymous dataset (see the link below). Additionally, **two co-authors** conducted the final verification for all subtasks.
>
> **Annotation Pipeline for Other Subtasks:** Thanks for the catch! All subtasks involved human annotation as described in Appendix B. Since the annotation processes for the other subtasks were relatively straightforward, we did not provide extensive details. We will expand Section B to clarify the annotation processes for all tasks.
>
> **Open-Source Raw Data:** **Yes! We plan to make all resources open-source.** This includes URLs, access dates, raw crawled data, final data samples for different tasks, the code for rendering websites, and the inference code for evaluation. These resources have been uploaded to an anonymous repository for preview (https://anonymous.4open.science/r/Anonymous_WebBench-E7D4) and will be publicly released.

---

> > ### Comment · Reviewer_f5BP · 2024-06-04
> > **Post-Rebuttal Response**
> >
> > Thank you for the explanation of the ROUGE-L vs. F1 metrics; I think this is an important discussion, and I'm glad to see these results. I'm also very excited that data and rendering code will be open-sourced!
> >
> > This was a very strong paper to begin with, and my opinion hasn't changed; I definitely will be pushing for acceptance at this time!

---

> > > ### Author Response · Authors · 2024-06-04
> > > **Thank you!**
> > >
> > > Thank you for your positive feedback and enthusiasm about the paper! Your strong support for accepting the paper is much appreciated. Thank you again for taking the time to review the paper thoroughly and provide constructive comments!
> > >
> > > Best,
> > > Authors

---

### Decision · Program_Chairs · 2024-07-10

**Decision:**

Accept

**Comment:**

This paper introduces a new benchmark for evaluating multimodal language models. All reviewers found the paper to be useful and of high quality. The authors are highly encouraged to implement the valuable feedback given by the reviewers since this will likely contribute to further improve the paper.